# K-LITE: Learning Transferable Visual Models with External Knowledge

**Sheng Shen**[*♮], **Chunyuan Li**[*†♠], **Xiaowei Hu**[*†], **Jianwei Yang**[†], **Yujia Xie**[†],
**Pengchuan Zhang**[†], **Zhe Gan**[†], **Lijuan Wang**[†], **Lu Yuan**[†]
**Ce Liu**[†], **Kurt Keutzer**[♮], **Trevor Darrell**[♮], **Anna Rohrbach**[♮], **Jianfeng Gao**[†]
[†]Microsoft    [♮]University of California, Berkeley

## Abstract

The new generation of state-of-the-art computer vision systems are trained from natural language supervision, ranging from simple object category names to descriptive captions. This form of supervision ensures high generality and usability of the learned visual models, due to the broad concept coverage achieved via large-scale data collection process. Alternatively, we argue that learning with *external knowledge* is a promising way which leverages a much more structured source of supervision and offers sample efficiency. We propose K-LITE[1], a simple strategy to leverage external knowledge for building transferable visual systems: In training, it enriches entities in text with WordNet and Wiktionary knowledge, leading to an efficient and scalable approach to learning image representations that uses knowledge about the visual concepts. In evaluation, the text is also augmented with external knowledge and then used to reference learned visual concepts (or describe new ones) to enable zero-shot and few-shot transfer of the pre-trained models. We study the performance of K-LITE on two important computer vision problems, image classification and object detection, benchmarking on 20 and 13 different existing datasets, respectively. The proposed knowledge-augmented models show significant improvement in transfer learning performance over existing methods. [2]

## 1 Introduction

One of the core aspirations in computer vision (CV) is to develop systems that endow computers with the ability to effectively learn general visual representations, which can be transferred to a variety of downstream recognition datasets with arbitrary visual concepts in the wild. Though excellent performance has been achieved on standard benchmarks, the traditional supervised approaches are limited to learning a fixed set of concepts, *e.g.,* 22K concepts on ImageNet [15] or 18K concepts on JFT-300M [82]. This leads to a few issues: $(i)$ Annotating each individual vision dataset is not only labor intensive, but also results in a narrow set of visual concepts; $(ii)$ Visual models trained on such datasets are good at one task (with the given concept set) and this task only, and show poor transfer learning performance to customized datasets that usually come with a different set of concepts [27].

To tackle this problem, recent large-scale language-augmented visual models, such as CLIP [71], ALIGN [36] and Florence [101], are trained on a wide variety of images with natural language supervision that is abundantly available on the Internet. These models demonstrate strong zero-shot transfer capabilities, since they acquire open-set recognition abilities through problem reformulation from classification to retrieval. Moreover, model generalization is improved as natural language

---

[*]Equal Technical Contribution    [♠]Project Lead
[1]**K**nowledge-augmented **L**anguage **I**mage **T**raining and **E**valuation
[2]Our code is available at `https://github.com/microsoft/klite`.

36th Conference on Neural Information Processing Systems (NeurIPS 2022).

supervision typically contains rich semantics. While these models usually perform well on recognizing common objects, they still struggle on visual concepts that are absent or rare in the pre-training stage. To ensure good transfer performance, it is required to train such models on huge datasets with sufficient concept coverage(*e.g.,* >400M image-text pairs), which is both labor and compute expensive.

Instead of scaling the number of image-text pairs to increase concept coverage, we propose to leverage *structured external knowledge* to augment language supervision. The inspiration comes from how humans generalize to novel concepts: instead of trying to memorize all concepts, humans leverage the structured knowledge such as definitions and concept hierarchy. For example, when we visit a Japanese restaurant for the first time, we may struggle to understand the menu by only looking at the dish names (*e.g.,* `Takoyaki, Sashimi`), as it is hard to imagine what they are. However, it becomes much clearer once a waiter introduces these concepts (Figure 1), leading to success in ordering food (*i.e.,* matching content to a name). Similar intuitions have been exploited in computer vision for class-level transfer [94, 12], but not yet for task-level transfer settings (similar to that of CLIP).

To this end, we explore a systematic approach to acquire and learn with external knowledge sources from databases such as WordNet [63] and Wiktionary [62] to train more transferable and sample-efficient visual models. The concept descriptions and concept hierarchies are purely textual, and the process of collecting external knowledge is fully automatic without extra human annotation. The acquired knowledge typically provides information that is shared between seen and unseen concepts to facilitate effective transfer. Specifically, rare concepts, *e.g.,* `Takoyaki, Sashimi` in Figure 1, are explained with more common concepts. Such knowledge sources are generally available for a variety of domains and datasets, making it possible to build a generic approach for task-level transfer.

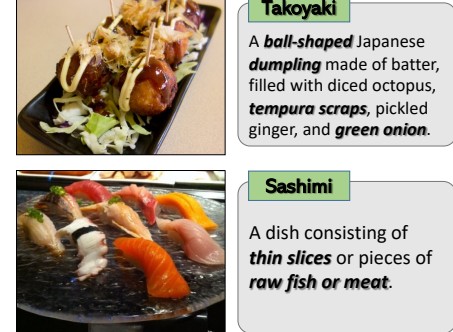

Figure 1: Motivating examples: knowledge explains the content of the rare dish concepts.

Our main findings and contributions can be summarized as follows:

- We present the first strong evidence that external knowledge can benefit *large-scale task-level transfer* for two core CV problems, image classification (IC) and object detection (OD), by exploring external knowledge sources, including WordNet and Wiktionary.

- A simple and effective strategy K-LITE (Knowledge-augmented Language Image Training and Evaluation) is proposed: The acquired knowledge is appended to the original textual concepts as model input during pre-training and evaluation. It can be viewed as an automatic knowledge-aware language prompting, which makes it easier for the model to access relevant information shared between the training and evaluation data. A modularized approach is also developed to enable efficient adaptation from vanilla visual models to their knowledge-augmented versions.

- To demonstrate the generality of the K-LITE, we instantiate it with two recent visual models and develop our knowledge-augmented counterparts: UniCL [95] for IC and GLIP [50] for OD. Extensive experiments in zero-shot and few-shot learning settings demonstrate that knowledge-augmented models can significantly improve over prior work. Notably, our model can achieve similar zero-shot performance to previous methods using only half of pre-training image-text pairs in some scenarios, demonstrating sample efficiency of the proposed approach.

## 2 Related Work

**Zero-shot Visual Recognition:** Zero-shot learning, *i.e.,* classifying images where there is a lack of labeled training data, has been studied for decades [21], and its popularity has recently increased further [94]. Based on the technique evolution, it can be broadly categorized into two generations: the traditional *class-level* zero-shot and recently popular *task-level* zero-shot setting.

*Class-level Transfer.* Class-level zero-shot learning aims to recognize object classes whose instances have not been observed during training. The goal for the zero-shot learning methods is to associate observed and non-observed classes through some form of auxiliary information, which can be either implicit such as pre-trained semantic embeddings [92, 78, 10], or explicit such as attributes [21, 43,

[35], text [18, 19, 72, 70], knowledge graphs [91, 74], or rules and ontologies [23]. Please, refer to the recent survey on knowledge-aware zero-shot learning for a more detailed review [12]. Recently, it has been suggested to move away from the restricted nature of standard zero-shot evaluation and make the task more practical by including training classes at test time, *i.e., generalized* zero-shot learning setting [94]. Despite the progress in this area, the traditional setting is typically limited to studying zero-shot transfer across classes in a *single domain* with manually defined splits, such as Animal with Attributes (AwA) [44], Birds-200 [89], SUN attributes [67], and ZS-ImageNet [73, 26]. Concurrently, [85] explore leveraging external knowledge to improve long-tailed visual recognition within individual domains, which falls into the category of class-level transfer.

*Task-level Transfer.* Another line of work focuses on task-level zero-shot transfer [52, 46, 64, 71, 36, 97]. They pre-train visual models on hundreds of millions of web-crawled image-caption or image-tags pairs, and evaluate their transfer ability by directly performing inference in a wide range of downstream datasets, without tuning the model weights. We argue that the task-level transfer is more practical and attractive than class-level transfer, as it is more relevant to real-world scenarios, where we may want to develop models that can serve many visual recognition applications.

Our work bridges the gap between the two lines of works above: it borrows the spirit of exploring knowledge in class-level transfer, and generalizes it for task-level transfer, leveraging the best of both worlds. To summarize, our work is different in two major aspects: ($i$) *Settings.* We focus on the task-level transfer learning across domains, *i.e.,* from large publicly available datasets to a diverse set of downstream datasets in different domains, and demonstrate that external knowledge benefits task-level transfer. ($ii$) *Modeling.* Existing class-level transfer works are built upon pre-trained visual features/backbones and shallow word embeddings or tf-idf scores, and only train the classifiers. One representative example is DeViSE [25], where a skip-gram word embedding model and an image classifier are fine-tuned jointly. In contrast, we are training large Transformer-based models in an end-to-end manner from scratch as in CLIP/ALIGN, providing the first empirical evidence that external knowledge can help train a general visual backbone.

**Knowledge-Intensive Models:** In natural language processing (NLP), with the increase of model capacity via pre-trained language models [17], there emerges the need for more knowledgeable models [55] with advanced functionalities such as making use of encyclopedic [88, 4, 6] and commonsense knowledge [79, 102]. To address this, a large number of language models augmented with external knowledge sources have been proposed [68, 29, 45, 56, 100, 7], achieving strong performance on a variety of NLP tasks [69, 42]. Please refer to a recent survey [98] for a comprehensive review.

In vision-and-language (V+L) domain, researchers have also started exploring knowledge-intensive tasks, e.g., OK-VQA [61] and WebQA [9]. They often require additional information sources (*e.g.,* factual and commonsense knowledge) beyond the QA pairs, compared to the established tasks such as VQA [3, 34] and image captioning [54, 2]. Hence, existing pre-trained models [49, 83, 58, 51, 81, 103, 38, 48, 30, 99, 77] would perform poorly on these knowledge-intensive V+L tasks [9]. To address the problem, acquiring external knowledge becomes an essential component for success [93, 60, 96].

The success of knowledge in NLP and V+L tasks inspires us to ask a natural question: Can we learn a transferable visual backbone model with external knowledge? Thus, we dissect and borrow the vital elements such as knowledge sources [63, 62] and modeling techniques [90, 100], and carry out studies for core computer vision tasks.

## 3 Knowledge-Augmented Visual Models

**Problem setup.** Computer vision systems have achieved strong transfer performance, when learning with large-scale image-label data [39] and image-caption data [71]. Recently, it has been demonstrated in [95, 101] that the unification of image-label and image-text formats into image-text-label achieves superior performance over either of them. We follow the setting in [95], and define a unified triplet-wise data format $\mathcal{D} = \{(\boldsymbol{x}_n, \boldsymbol{t}_n, y_n)\}_{n=1}^N$, where $\boldsymbol{x} \in \mathcal{X}$ is an image, $\boldsymbol{t} \in \mathcal{T}$ is its language description, and $y \in \mathcal{Y}$ is a label indicating the index of the unique language description in the dataset. In a general form, the language description is a text sequence $\boldsymbol{t} = [t_1, \cdots, t_L]$. It ranges from simple category names representing visual concepts when $L$ is small, to more free-form and semantic-rich sentences such as captions when $L$ is relatively large.

In this paper, we assume there exists an external knowledge source $\mathcal{S}$, where one may use the language description $\boldsymbol{t}$ as a query to seek additional knowledge description $\boldsymbol{s} \in \mathcal{S}$ for $\boldsymbol{t}$. Given these triplet data instances $\mathcal{D}$ and an external knowledge source $\mathcal{S}$, our goal is to learn generic visual-

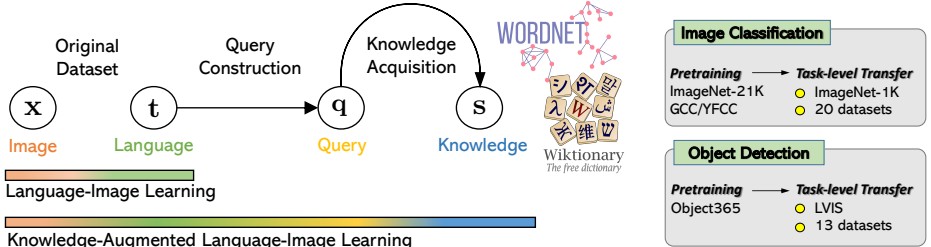

Figure 2: Left: Illustration of data construction process of the proposed knowledge-augmented language-image learning, in contrast to the baseline language-image learning. The query $q$ is constructed from Eq.(1). The same process is performed for both pre-training and downstream tasks. Right: The proposed strategy is applied to IC and OD for task-level transfer.

semantic representations, which are readily transferable to a wide range of downstream datasets, whose category names are not necessarily observed during training. In Figure 2, we visually illustrate the proposed knowledge-augmentation process and two considered application scenarios.

## 3.1 External Knowledge

**Query Construction.** For a text sequence associated with an image, different tokens may play different roles in contributing to describing the main semantics of the image. Humans leverage this prior inherently in parsing the sentences to understand the image. Further, it is infeasible to employ the entire sequence as a query, as this may lead to the lack of coverage in the knowledge bases. Instead, we propose to construct a query $q \in \mathcal{Q}$ as a compact form of original language description $t$, represented with the words that convey the main concepts of the image.

Specifically, we consider a divide-and-conquer approach. For short text sequences $t$ such as category names in image-tag/label datasets (*e.g.,* ImageNet [15]), we directly use the category name as the query. For long text sequences $t$ such as captions (*e.g.,* YFCC [84]), we first parse the sentence to extract the noun-phrases, among which the most rare noun-phrase over the corpus is used as a query for this sentence. The intuition is to convert the rare concepts into "explanations" represented in common words using external knowledge. Noun phrases are useful for summarizing the sentence and thus inferring what is being talked about in the image. For example, in "`professional boxer is introduced to the crowd`", the noun-phrases are "`boxer`", "`professional boxer`", and "`the crowd`". We summarize the query construction process $g_{query}$ below for clarity:

$$q = g_{query}(t) = \begin{cases} t, & \text{when } t \text{ is a category, class or tag name,} \\ \text{The most rare noun-phrase}(t), & \text{when } t \text{ is a caption.} \end{cases} \quad (1)$$

**Knowledge Acquisition from External Sources.** We consider three knowledge sources $\mathcal{S}$ to enrich the language descriptions $t$. They are constructed based on the two knowledge bases: WordNet [63] and Wiktionary [62]. To measure the breadth of a knowledge base, we define the *concept coverage* as the percentage of non-empty knowledge items retrieved for a given set of issued queries.$(i)$ WordNet [63] is a lexical database which links words into semantic relations including synonyms, hyponyms, and meronyms. Both nouns and verbs are organized into hierarchies, defined by hypernym relationships. The synonyms in WordNet are grouped into *synsets*, expressing the same distinct concept. The synsets serve as a natural link between language and vision domains. For example, ImageNet is an image database organized according to the WordNet hierarchy, where each node is depicted by hundreds/thousands of images. $(ii)$ Wiktionary [62] is a web-based content dictionary of terms (including words, phrases, proverbs, linguistic reconstructions). These entries may contain definitions, illustrations, usage examples *etc.*. Next, we provide our knowledge retrieval process $s = g_{retrieve}(q)$ for each source $\mathcal{S}$, followed by an example result for the query "`boxer`":

- *WordNet Hierarchy* $\mathcal{S}_{\text{wn\_path}}$. A WordNet node of the query is located, then we repeatedly search its parent node. The words along the traversal path are recorded as the knowledge.
  - `[boxer, combatant, person, causal_agent, physical_entity, entity]`
- *WordNet Definition* $\mathcal{S}_{\text{wn\_def}}$. The definition from the synsets is used to explain the query.
  - `someone who fights with his fists for sport`
- *Wiktionary Definition* $\mathcal{S}_{\text{wiki\_defh}}$. The query is used for dictionary look-up in Wiktionary, and the corresponding definition is used.
  - `a fighter in a boxing match`

When multiple meanings (senses) exist for a given query, we simply consider the first one for simplicity, and leave more sophisticated designs as future work. After querying each knowledge source, we represent the external knowledge for each language description $t$ in the form of concatenation of its query and the corresponding retrieved result: $[q, s]$. This external knowledge introduces additional supervision signals to guide visual models to learn better aligned visual-semantic representations, as we explain later. We next describe how to encode knowledge in multimodal models to improve transfer in the image-level task of image classification and the region-level task of object detection.

## 3.2 Image Classification

Recent works that learn visual models with language supervision [71] often employ a dual-encoder architecture. For each image $x$, an image encoder model $f_{\theta}$ parameterized by $\theta$ first represents $x$ as a visual feature vector $\tilde{v} \in \mathbb{R}^{P \times 1}$: $\tilde{v} = f_{\theta}(x)$. For each language description $t \in \mathcal{T}$, we encode it with a text encoder $f_{\phi}(t)$ parameterized by $\phi$, and get the `[EOS]` feature as the vector representation of the sentence $\tilde{u} \in \mathbb{R}^{P \times 1}$: $\tilde{u} = f_{\phi}(t)$. In this paper, we further leverage this text encoder to encode the external knowledge. First, the query $q$ is represented in natural language $p = g_{prompt}(q)$ using the language prompt as in [71]. Depending on the language input $t$ and our augmentation scheme, the knowledge-augmented text sequence $t^k \in \mathcal{T}^k$ ($k$ stands for *k*nowledge) is represented as:

$$t^k = \begin{cases} t_e^k = [p, q, s], & \text{when } t \text{ is a category, class or tag name,} \\ t_c^k = [t, q, s], & \text{when } t \text{ is a caption, and a } \texttt{concat} \text{ scheme is used,} \\ \{t_c^k, t_e^k\}, & \text{when } t \text{ is a caption, and a } \texttt{combine} \text{ scheme is used.} \end{cases} \quad (2)$$

For example, for category name $t = $ `boxer` or for caption $t = $ `professional boxer is introduced to the crowd`, we have $q = $ `boxer`, and the corresponding language description are:

- $t_e^k = $ `a photo of a cool boxer ; boxer , a fighter in a boxing match`
- $t_c^k = $ `professional boxer is introduced to the crowd ; boxer , a fighter in a boxing match`

We re-use the same text encoder $f_{\phi}$ to encode $t^k$ as the original $t$ as supervision for image $x$.

**Training.** For $i$-th image $x_i$ and $j$-th language description $t_j$ in a batch $\mathcal{B}$, we normalize their feature vectors in a hyper-sphere using $u_i = \frac{f_{\theta}(x_i)}{\|f_{\theta}(x_i)\|}$ and $v_j = \frac{f_{\phi}(t_j)}{\|f_{\phi}(t_j)\|}$, and their similarity is calculated as $u_i^{\top} v_j$. A bidirectional supervised contrastive objective is considered to train the model:

$$\min_{\{\theta, \phi\}} \mathcal{L}_{\text{IC}} = \mathcal{L}_{i2t} + \mathcal{L}_{t2i}, \text{ with} \quad (3)$$

$$\mathcal{L}_{i2t} = -\sum_{i \in \mathcal{B}} \frac{1}{|\mathcal{P}(i)|} \sum_{k \in \mathcal{P}(i)} \log \frac{\exp(\tau u_i^{\top} v_k)}{\sum_{j \in \mathcal{B}} \exp(\tau u_i^{\top} v_j)} \text{ and } \mathcal{L}_{t2i} = -\sum_{j \in \mathcal{B}} \frac{1}{|\mathcal{Q}(j)|} \sum_{k \in \mathcal{Q}(j)} \log \frac{\exp(\tau u_k^{\top} v_j)}{\sum_{i \in \mathcal{B}} \exp(\tau u_i^{\top} v_j)}$$

where $\mathcal{P}(i) = \{k | k \in \mathcal{B}, y_k = y_i\}$, $\mathcal{Q}(j) = \{k | k \in \mathcal{B}, y_k = y_j\}$, and $\tau$ is a temperature hyper-parameter controlling the strength of penalties on hard negative samples. Note (3) is a general form; it reduces to the training objective of CLIP [71] or ALIGN [36] when there is a one-to-one mapping between an image and its paired caption in a batch, *i.e.,* $\mathcal{P}(i) = \{i\}$ and $\mathcal{Q}(j) = \{j\}$.

**Evaluation.** Given a downstream image classification task with a custom set of category names, we represent them with the knowledge-augmented prompt form in (2); they are fed into the pre-trained text encoder $f_{\phi}$ to obtain the class embedding. The test image $x$ is encoded with $f_{\theta}(x)$, and compared to all class embeddings to get its label from the best matching class.

**Extensions with Modularized Modeling.** In our study, we found it is key to ensure consistency between training and evaluation stages: if a model is trained with knowledge, it performs well when

also evaluated with knowledge. Similarly, if a model is trained without knowledge (*e.g.,* CLIP/UniCL), adding knowledge directly in the evaluation stage results in performance drop. However, due to the limited knowledge coverage in existing knowledge bases, $s$ could be empty for a large number of queries $q$. When the low coverage happens for a downstream evaluation dataset, it may result in a training-evaluation inconsistency for our knowledge-augmented models, and thus lower performance.

It is desired to have a modularized model that can switch between "with" and "without" knowledge settings. Inspired by [90], we propose to employ adapters [33] to build the network branch to encode knowledge-augmented language $t^k$, where serial MLP adapters are inserted after each self-attention and MLP modules for all Transformer layers of the text encoder, and $t^k$ is passed through $f_\phi$ and adapters. Meanwhile, the original $f_\phi$ is reserved as the branch to encode vanilla natural language $t$. The proposed adapter-modularized architecture can also be used for efficient stage-wise continual pre-training: one may start with a vanilla language-image model pre-trained on $\mathcal{D}$, and continue pre-train the adapters with knowledge-augmented data $(\mathcal{D}, \mathcal{S})$ to build its knowledge version.

### 3.3 Object Detection

Object detection (OD) typically involves two tasks: $\mathcal{L}_{OD} = \mathcal{L}_{cls} + \mathcal{L}_{loc}$, where the localization task $\mathcal{L}_{loc}$ aims to locate the presence of objects in an image with a bounding box, and classification task $\mathcal{L}_{cls}$ determines what object categories are present in that box. Similar to the IC task above, we improve the categorization task of individual boxes $\mathcal{L}_{cls}$ in OD with external knowledge, and keep $\mathcal{L}_{loc}$ the same. Specifically, we leverage GLIP [50] to reformulate OD as a phrase grounding task, by grounding each region proposed by $\mathcal{L}_{loc}$ [53] to phrases in a text sequence. For language encoding, we first augment a category name $t$ into its knowledge-augmented form $t^k = [q, s]$; this is different from IC in that $p$ is excluded, as no prompt engineering is used in OD, as in [50]. In the original GLIP, a *sequential text encoding* scheme is used: a concatenated long sequence $[q_1, \cdots, q_K]$ over category names is considered as the text encoder input. In our case, simple concatenation $[q_1, s_1 \cdots, q_K, s_K]$ will quickly break the max length requirement of the language encoder.

To resolve the issue, we propose a *parallel text encoding* scheme: each $t^k$ is passed through language encoder independently, we use the top-layer feature of [CLS] token as the contextual vector representation $\tilde{u} \in \mathbb{R}^{P \times 1}$ of $t^k$: $\tilde{u} = f_\phi(t^k)$. Given $K$ categories, they can be encoded in parallel in a batch; the encoded phrase feature sequence is the concatenation $\mathbf{U} \in \mathbb{R}^{P \times K}$: $\mathbf{U} = [\tilde{u}_1, \cdots, \tilde{u}_K]$. The region encoding is the same as in GLIP. The feature pyramid is $\mathbf{V} \in \mathbb{R}^{M \times P}$ : $\mathbf{V} = f_\theta(x)$, where $M$ is the number of box features. The alignment scores $\mathbf{S}_{\text{ground}}$ are computed:

$$\mathbf{S}_{\text{ground}} = \mathbf{V}\mathbf{U}, \quad \mathcal{L}_{cls} = \mathcal{M}(\mathbf{S}_{\text{ground}}; \mathbf{T}), \tag{4}$$

where $\mathbf{T} \in \{0, 1\}^{M \times K}$ is the target indicating match or no match, and $\mathcal{M}(\mathbf{S}; \mathbf{T})$ is the focal loss [53]. The grounding model, consisting of the image encoder $f_\theta$, the language encoder $f_\phi$ and a cross-modal interaction head introduced in [50], is trained end-to-end by minimizing the loss defined in (4). In the evaluation stage, the external knowledge is also retrieved, and encoded in the same parallel encoding manner to enrich the category names in the downstream OD tasks.

## 4 Experimental Results

In this section, we examine our knowledge-augmented approach to answer two research questions. Q1: To what extent external knowledge benefits visual transfer learning, including sample-efficiency in pre-training and downstream? Q2: Why does external knowledge help zero-shot transfer (illustrated with success and failure case studies)?

### 4.1 Settings

**Evaluation benchmark.** We apply the proposed knowledge-augmented models to two task-level transfer settings defined in ELEVATER benchmark [47], which evaluates the transferability of the learned visual representations in the wild. We study our models based on the datasets described in Table 1. The license, PII, and consent details of each dataset are in the respective papers. Due to the limited computational resources, the pre-training datasets are constrained to the large publicly available datasets used in [95, 50]. This setting is defined as the "Academic Track" in [47], which friendly to the academic community to allow reproducibility of the results. The number of visual concepts is identical to the number of categories for datasets with category names (*e.g.,* ImageNet

| Task | | Pre-training | | | | Downstream Concept Overlap (%) | |
|---|---|---|---|---|---|---|---|
| | | #Instances | #Concepts | Vocab. Size | #Ins/#C. | | |
| | Dataset | | | | | ImageNet-1K | 20-datasets |
| IC | ImageNet-21K [15] | 13M | 19.2K / 18.4K | 13.5K / 12.9K | $591 \pm 537$ | 11.82 | 13.26 |
| | GCC-3M [76] | 3.3M | 681K / 64.5K | 29.6K / 13.0K | $9.5 \pm 303$ | 35.97 | 19.73 |
| | GCC-12M [11] | 12M | 10.2M / 728K | 1.24M / 264K | $5.6 \pm 353$ | 61.02 | 31.34 |
| | YFCC-14M [84] | 14M | 14.2M / 1.25M | 2.41M / 473K | $8.3 \pm 1354$ | 65.23 | 34.65 |
| OD | Dataset | | | | | LVIS | 13-datasets |
| | Object-365 [75] | 9.6M | 365 / 365 | 452 / 452 | $26.3K \pm 12.4K$ | 13.46 | 21.26 |

Table 1: Statistics of training and test datasets used in our experiments. #Instances indicates #Image for IC and #Regions for OD, respectively. For #Concept and Vocabulary size, we report numbers for the full set and for items with frequency larger than 5. #Ins/C. reports the mean and standard derivation for the numbers of instances per concept.

| Training Method | Knowledge $\mathcal{S}$ | ImageNet-1K | | ICinW (20 datasets) | |
|---|---|---|---|---|---|
| 1-branch, from scratch | - | 28.16 | 4.93 | 27.15 | 17.10 |
| | $\mathcal{S}_{\text{wn\_hier}}$ | 27.43 | 29.03 | 28.15 | 28.69 |
| | $\mathcal{S}_{\text{wn\_def}}$ | 22.87 | 29.31 | 26.97 | 29.14 |
| | $\mathcal{S}_{\text{wiki\_def}}$ | 22.05 | **30.23** | 29.03 | **33.44** |
| 2-branch, continue pre-training | $\mathcal{S}_{\text{wiki\_def}}$ | 28.16 | 28.40/28.90 | 27.15 | 30.73/30.91 |
| 2-branch, from scratch | $\mathcal{S}_{\text{wiki\_def}}$ | 28.16 | 32.52/32.44 | 27.15 | **32.46/33.49** |

Table 2: Zero-shot task transfer performance after pre-training on ImageNet-21K dataset. The top block studies the effectiveness of knowledge sources $\mathcal{S}$, and the bottom block studies the modularized approach. For each downstream task, the 1st and 2nd column reports the results without and with knowledge, namely green cells indicate "a match", orange cells indicate "a mismatch" w.r.t. adding knowledge in training and evaluation. In the bottom block, we report two numbers when evaluated with knowledge: using the knowledge branch only, and using two branches selectively.

and Object-365). For image-text data (bottom 3 rows of the IC block), we use Spacy [32] to extract the noun phrases. We also use a merged version of GCC-3M and GCC-12M denoted as GCC-15M. Given the pool of concepts, we calculate the number of unique words and report it as the vocabulary size. For Concepts and Vocab Size, we report 2 numbers: first for the full set, second for items with frequency larger than 5. The latter provides a sense of "long-tailness". The statistics (*e.g.,* ratio of #Instance / #Concept) illustrates the varied trade-off over different datasets: image diversity, semantic richness and long-tailness. For example, YFCC is the most long-tail dataset in IC, as it has low mean value and the largest standard derivation value in #Instance / #Concept. The *concept overlap* is computed as the percentage of concepts in a downstream dataset that are covered by the pre-training dataset. For 20-datasets and 13-datasets, the averaged overlap across individual datasets is reported. It measures the gap (or difficulty) in concept transfer between the pre-training and the downstream data. The dataset statistics are detailed in Section B.1 in Appendix.

*Zero/Few-shot image classification.* Following UniCL [95], this task evaluates to what extent a model understands novel concepts. We pre-train on ImageNet-21K [15] and GCC [76, 11]/YFCC [84] datasets, and report results on ImageNet-1K [15] and a suite of 20 datasets (ICinW) proposed in [47]. We use the same text prompts as in [71, 95], and report `scores` averaged over 20 datasets. UniCL/Florence [101] show superior performance to CLIP or ALIGN counterparts; UniCL is Florence in a controlled academic setting, trained on the large publicly available datasets [95].

*Zero-shot object detection.* Following GLIP [50], we pre-train on Object365 [75], and transfer the learned visual representations for object detection on LVIS [28] and a suite of 13 small OD datasets (ODinW) proposed in [50, 47], to check the generalization ability. The box `mAP` is reported.

## 4.2 Image Classification

**ImageNet-21K pre-training.** We start by pre-training on ImageNet-21K, where all ImageNet-1K images are excluded. We still see a small amount of concept overlap in Table 1, and hypothesize that some category names are given based on different level of WordNet hierarchy. The benefits of this setting are two-fold: it ensures distinctively less concept overlap, and all concepts can find their full WordNet knowledge. We report the results in Table 2. For each checkpoint, we report the results without and with knowledge in the evaluation stage. We confirm two major findings below.

| Training Data | | Method | ImageNet-1K | ICinW (20 datasets) | | |
|---|---|---|---|---|---|---|
| Dataset | # Samples | | Zero-shot | Zero-shot | Linear Probing | Fine-tuning |
| ImageNet-21K | 13M (full) | UniCL | 28.16 | 27.15 | $53.07 \pm 4.15$ | $55.96 \pm 2.50$ |
| | 13M (full) | K-Lite | **30.23** | **33.44** | $\mathbf{53.92} \pm \mathbf{1.05}$ | $\mathbf{57.81} \pm \mathbf{1.48}$ |
| YFCC-14M + ImageNet-21K | 14M (half) | UniCL | 34.43 | 34.30 | $53.50 \pm 2.22$ | $56.45 \pm 2.48$ |
| | 14M (half) | K-Lite | 36.67 | 36.50 | $49.48 \pm 2.23$ | $55.88 \pm 1.64$ |
| | 14M (half) | K-Lite$^\diamond$ | 42.36 | 36.50 | $54.28 \pm 3.66$ | $52.11 \pm 4.90$ |
| | 27M (full) | UniCL | 43.06 | 35.99 | $55.96 \pm 3.38$ | $58.25 \pm 2.98$ |
| | 27M (full) | K-Lite | **45.67** | **38.89** | $\mathbf{57.06} \pm \mathbf{1.48}$ | $58.24 \pm 2.36$ |
| GCC-15M + ImageNet-21K | 15M (half) | UniCL | 41.64 | 36.31 | $53.86 \pm 2.73$ | $59.04 \pm 3.13$ |
| | 15M (half) | K-Lite | 44.26 | 39.53 | $55.91 \pm 2.53$ | $58.20 \pm 3.39$ |
| | 15M (half) | K-Lite$^\diamond$ | 47.30 | 40.32 | $57.38 \pm 2.70$ | $60.72 \pm 2.29$ |
| | 28M (full) | UniCL | 46.83 | 38.90 | $57.92 \pm 3.31$ | $60.99 \pm 2.74$ |
| | 28M (full) | K-Lite | **48.76** | **41.34** | $\mathbf{58.56} \pm \mathbf{3.12}$ | $\mathbf{63.39} \pm \mathbf{1.74}$ |

Table 3: Overall comparisons of our knowledge-augmented models. Each model is pre-trained with 32 epochs following CLIP [71]/UniCL [95]. $^\diamond$ It indicates that the `Combine` scheme is used for the image-caption data, otherwise the default is the `Concat` scheme decribed in Section 3.2. The linear probing and fine-tuning results are reported for 5-shot settings over 3 random seeds.

*F1: All three knowledge sources are beneficial.* In Section 3.1, we have introduced WordNet hierarchy $\mathcal{S}_{\text{wn\_path}}$, WordNet definition $\mathcal{S}_{\text{wn\_def}}$, and Wiktionary definition $\mathcal{S}_{\text{wiki\_def}}$ as external knowledge sources. It is shown that all three of them are effective, improving the zero-shot accuracy by absolute gain 1-2% on ImageNet-1K (from 28.16% to 30.23%) and 2-6% on the dataset suite in average (from 27.15% to 33.44%), respectively. Among them, Wiktionary definition $\mathcal{S}_{\text{wiki\_def}}$ turns out to be the most effective, therefore, we use it as the default knowledge source throughout the remaining experiments.

*F2: The modularized approach is effective.* Our results in Table 2 reveal that training/test inconsistency in terms of involving knowledge can dramatically degrade the model performance. For example in the 1st row, the baseline UniCL is pre-trained without knowledge, its performance decreases from 28.16% to 4.93% when knowledge is added in the test stage. In contrast, in the 4th row, our K-Lite is pre-trained with knowledge, its performance decrease from 33.44% to 29.03% if knowledge is excluded in the evaluation stage. Therefore, we consider a modularized approach with 2-branch in the model. First, we continue pre-train our modularized model from a 32-epoch knowledge-free checkpoint by only updating the Adapters on knowledge-augmented image-text pairs for 10 epochs. It already shows a performance gain from 27.61% to 28.40%. This suggests a more affordable solution to obtain knowledge-augmented models from existing models. We can further boost the performance to 28.90% if we evaluate with two branches, each of which only passes its corresponding language version. Finally, we also train the modularized model from scratch, and it demonstrates a significant gain (absolute 4%) on ImageNet-1K, and over 5% improvement on the 20 datasets. For fair comparisons with knowledge-free models, we train our models with one branch in the rest of experiments.

To demonstrate the performance of K-Lite in the extreme large-scale settings, we leverage the largest checkpoint of Florence [101] trained on 800M image-text pairs, and continue pre-training the model on ImageNet-21K with external knowledge. It improves the zero-shot ImageNet-1K accuracy of Florence from 83.74% to 85.80%. As an ablation baseline, continuing pre-training without knowledge yields 85.35%. The absolute 0.45% performance gain shows that external knowledge can still benefit transfer learning, though a huge amount of pre-training data is employed.

**Pre-training on image-text-label data.** The unification of image-label and image-caption as image-text-label has been demonstrated superior over either one of them [95]. Therefore, we report overall results on the combined data in Table 3. The few-shot learning results are reported with 5 training examples, using two model adaptation method: linear probing and full model fine-tuning. The average numbers over 3 random seeds are reported. K-Lite improves its knowledge-free counterpart UniCL in almost all the cases. Importantly, K-Lite can outperform UniCL using only half of the pre-training image-text pairs in several cases. It demonstrate the high sample-efficiency of K-Lite, and that external knowledge is an effective source to consider, when collecting large-scale image-text pairs to develop language-augmented visual models at scale. We also compare K-Lite and UniCL with Swin-Base on the joint data including ImageNet-21K, GCC15M and YFCC15M. K-Lite

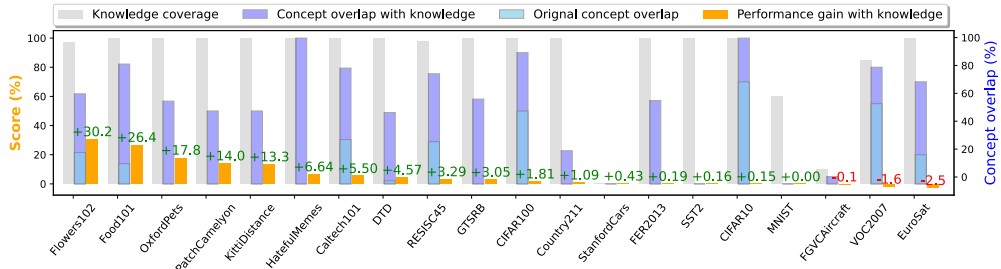

Figure 3: Performance improvement analysis with external knowledge. External knowledge can largely improve **concept overlap** between pre-training and evaluation stages, hence usually yields higher recognition **scores**. **Knowledge coverage** indicates the percentage of concepts that exist in the knowledge base for each downstream dataset.

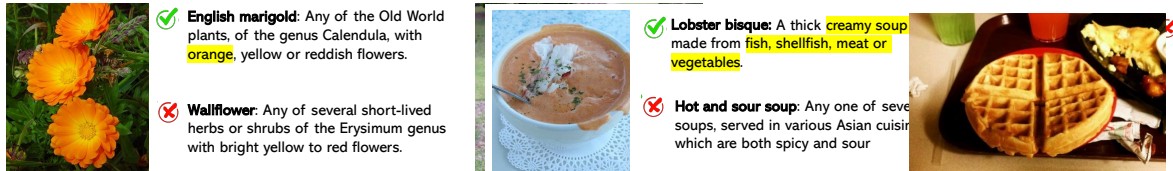

(a) Success examples. The two datasets with largest improvement in Fig. 3: Flowers102 and Food101. The description of the parent concept, material, shape, color *etc.* clarifies the concepts, boosting performance for the fine-grained classification tasks.

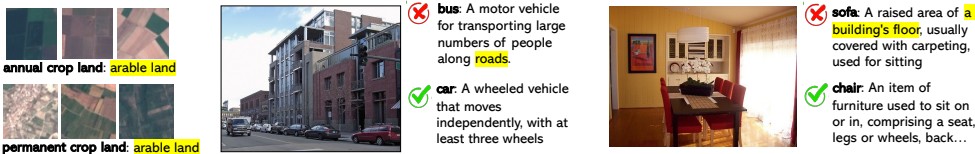

(b) Failure examples. The two datasets with the largest performance loss in Fig. 3. Left (EuroSat): both class names have the same knowledge. Middle & Right (VOC2007): The knowledge contains spurious words that confuse the models.

Figure 4: Success and failure cases on image classification. For each image, the top row is the knowledge-based prediction, and the bottom row is the baseline prediction (no knowledge).

improve the zero-shot performance of UniCL from 52.18% to 57.78% on ImageNet-1K, and from from 43.20% to 45.47% on ICinW.

**Breakdown Analysis.** Next, we ask why does external knowledge improve the zero-shot task transfer performance on a broad range of datasets? To answer this question, we compare the breakdown performance on all 20 dataset in Figure 3, for the ImageNet-21K checkpoints trained with and without Wiki knowledge. Out of 20 datasets, external knowledge shows superior/comparable/inferior performance to the baseline on 16/1/3 datasets, respectively. One prominent observation is that Wiki knowledge improves concept overlap for train-evaluation from 13.26% to 51.24% by average. It is easy to understand, concepts are explained in more commonly used words in Wiktionary, providing a bridge for train and evaluation. This is reflected by the increased height of blue bar for most datasets in Figure 3. Interestingly, for all datasets with increased accuracy scores, there shows an increase of the concept overlap. In summary, knowledge is an effective approach to improve concept overlap, a prerequisite for good task transfer performance. In Figure 4 (a), we provide success examples after adding knowledge; more examples are shown in Appendix.

**Limitations.** The failure cases where knowledge-augmented approach does not help mainly belong to two scenarios: ($i$) No external knowledge was extracted from the given knowledge base (*i.e.,* Wiktionary in this case), *e.g.,* StanfordCars and FGVC Aircraft. They often require domain-specific knowledge explanations to define a car brand (*e.g.,* Volvo C30 Hatchback 2012) or an aircraft model type (*e.g.,* 737-200), while Wiktionary can hardly provide such professional definitions. ($ii$) While knowledge is available, the quality is too low to provide useful information. In Figure 4 (b), we provide failure examples from two datasets with the biggest performance loss after adding

| Method | APr | APc | APf | LVIS | | | | | ODinW (13 datasets) | | | |
|---|---|---|---|---|---|---|---|---|---|---|---|---|
| | | | | - | $\mathcal{S}_{\text{LVIS}}$ | $\mathcal{S}_{\text{wn\_path}}$ | $\mathcal{S}_{\text{wn\_def}}$ | $\mathcal{S}_{\text{wiki\_def}}$ | - | $\mathcal{S}_{\text{wn\_path}}$ | $\mathcal{S}_{\text{wn\_def}}$ | $\mathcal{S}_{\text{wiki\_def}}$ |
| GLIP-A [50] | 14.2 | 13.9 | 23.4 | 18.5 | - | - | - | - | 28.8 | - | - | - |
| Baseline GLIP$^\heartsuit$ | 8.6 | 14.0 | 23.1 | 17.9 | 17.6 | 17.1 | 17.2 | 15.0 | 27.5 | 26.8 | 21.0 | 18.5 |
| K-LITE | 14.8 | 18.6 | 24.8 | 16.9 | 21.3 | 18.7 | **21.4** | 20.5 | 25.0 | 30.3 | 28.4 | **31.7** |

Table 4: Zero-shot task transfer performance on OD. APr/APc/APf indicates the AP values for rare, common, frequent groups of categories on LVIS. Cell coloring follows the same protocol as in Table 2. $^\heartsuit$GLIP is implemented with parallel text encoding in Section 3.3 without external knowledge.

knowledge. In Figure 6 in the Appendix, we show more examples where knowledge only yields slight improvement. To summarize, a promising future research direction is to improve the knowledge quality to be more related to the given classification tasks.

### 4.3 Object detection

We evaluate the model's ability to recognize diverse objects on LVIS [28] and 13 downstream datasets used in [50] in a zero-shot setting. We report on MiniVal containing 5,000 images on LVIS. Our K-LITE GLIP is trained with Wiktionary definitions. The results are presented in Table 4. The 1st row are the original numbers reported in [50], using the sequential text encoding. The 2nd row is our implementation of GLIP using parallel text encoding, whose effectiveness is validated by the comparable numbers with 1st row. The 3rd row is our knowledge-augmented GLIP, *i.e.,* K-LITE. The benefit of using external knowledge is evident. On LVIS, the categories are divided into rare, common, frequent groups, based on the number of training images per category. K-LITE improves the detection performance for all three groups with an average of 2.8 points on LVIS, and particularly brings a 4.7 points improvement on MiniVal APc over the GLIP-A reported in [50]. We conclude that the enriched semantics of external knowledge significantly helps the model recognize concepts with a decent number of instances. Since LVIS has its own knowledge source $\mathcal{S}_{\text{LVIS}}$, mostly built upon WordNet definitions [28], we evaluate our model with $\mathcal{S}_{\text{LVIS}}$. We alter the external knowledge source to $\mathcal{S}_{\text{wn\_path}}$, $\mathcal{S}_{\text{wn\_def}}$, and $\mathcal{S}_{\text{wiki\_def}}$ in evaluation, which yields mAP 18.7, 21.4, 20.5, respectively. It verifies that our knowledge source extraction process is reliable. Similarly, K-LITE improve the 13 downstream OD datasets from 28.8 (or 27.5 with its own knowledge-free counterpart) to 31.7. The success and failure examples of OD are shown in Section B.5 in Appendix.

## 5 Conclusions

In this paper, we have presented a knowledge-augmented approach K-LITE to learn a generic visual model for task-level transfer. General external knowledge sources including WordNet and Wiktionary are explored to enrich the natural language supervision, which is then used in both the language-image pre-training stage and the prompt-based evaluation stage. We have demonstrated the generality and effectiveness of K-LITE in two core computer vision problems: image classification and object detection. Extensive experimental results show that our method can achieve superior performance over existing methods on 20 IC datasets and 13 OD datasets, respectively. K-LITE also outperforms its knowledge-free counterpart UniCL using half of the pre-training data in the large-scale academic data setting, demonstrating that leveraging external knowledge is a promising direction in improving pre-training sample-efficiency for learning transferable visual models.

## Acknowledgments

The authors gratefully acknowledge Chenguang Zhu, Wenhao Yu, Yuwei Fang for the early insightful discussions on the use of dictionary for rare concepts in NLP task, Hao Cheng for the inspirations of external knowledge in open-domain QA. The project is partly done in the MSR-Berkeley collaboration program[3]. The work depends on publicly available knowledge databases; we acknowledge all the original authors and contributors who made their "knowledge" public. Sheng Shen and Kurt Keutzer are supported by Samsung SAIT, Intel corporation, Intel VLAB team, Intel One-API center of excellence, as well as funding through BDD and BAIR. The work of Sheng Shen, Anna Rohrbach and Trevor Darrell was supported in part by DoD including DARPA's LwLL, PTG and/or SemaFor programs, as well as BAIR's industrial alliance programs.

---

[3] https://www.microsoft.com/en-us/research/collaboration/bair

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
