# Supplementary Material for "K-LITE: Learning Transferable Visual Models with External Knowledge"

This appendix is organized as follows.

- In Section A (referred by CheckList), we discuss the societal impact.

- In Section B.1 (referred by Section 4.1), we summarize the statistics of the datasets used in in pre-training and evaluation stages.

- In Section B.2 (referred by Section 4), we introduce the pre-training and model adaptation details of our experiments on image classification (IC) and object detection (OD).

- In Section B.3, we provide zero-shot retrieval comparison by introducing knowledge.

- In Section B.4, we provide quantitative analysis on how external knowledge benefit transfer.

- In Section B.5 (referred by Section 4.2 and 4.3), we provide more visualizations of success and failure examples for K-LITE.

- In Section B.6, we provide more object detection results by training on larger dataset for K-LITE.

## A  Societal Impact

We do not anticipate a specific negative impact, but, as with any Machine Learning method, we recommend to exercise caution. The existing knowledge bases such as Word-Net and Wiktionary are the results of crowd-sourcing various human knowledge or commonsense into a centered place. K-LITE provides evidence to leverage such knowledge bases for AI research. It encourages the community to contribute more to improve the coverage and quality of knowledge items, which will further benefit AI research.

## B  Experimental Results

### B.1  Dataset statistics

In Table 5 and Table 6, we list the basic characteristics of the 20 IC datasets and the 13 OD datasets used in this paper, respectively.

### B.2  Training schedule

**UniCL for Image classification.**  Following UniCL, we use Swin-Tiny [57] as the visual encoder. For language encoder, we use a 12-layer Transformer [86] with hidden dimension of 512 following [71]. Features from visual and textual encoder are projected to the same dimension of 512, using two linear projection layers. All models including the baseline models are trained on 16 GPUs for 32 epochs, with batch size 4096, initial learning rate $1 \times 10^{-3}$ and weight decay 0.1. We also used a cosine learning rate scheduler with 5000 warmup iterations.

**GLIP for Object Detection.**  Following GLIP, we pre-train models based on Swin-Tiny models with 32 GPUs and a batch size of 64. We use a base learning rate of $1 \times 10^{-5}$ for the language backbone and $1 \times 10^{-4}$ for all other parameters. The learning rate is stepped down by a factor of 0.1 at the 67% and 89% of the total training steps.

**Adapting to the downstream tasks.**  To adapt the pre-trained model for downstream tasks on the evaluation benchmark, we follow the procedure in [47], where automatic hyper-parameters are

| Dataset | #Concepts | Train size | Test size | Evaluation metric | Source link |
|---|---|---|---|---|---|
| Hateful Memes [37] | 2 | 8,500 | 500 | ROC AUC | Facebook |
| PatchCamelyon [87] | 2 | 262,144 | 32,768 | Accuracy | Tensorflow |
| Rendered-SST2 [71] | 2 | 6,920 | 1,821 | Accuracy | OpenAI |
| KITTI Distance [24] | 4 | 6,347 | 711 | Accuracy | KITTI website |
| FER 2013 [1] | 7 | 28,709 | 3,589 | Accuracy | Kaggle fer2013 |
| CIFAR-10 [41] | 10 | 50,000 | 10,000 | Accuracy | Tensorflow |
| EuroSAT [31] | 10 | 5,000 | 5,000 | Accuracy | Tensorflow |
| MNIST [16] | 10 | 60,000 | 10,000 | Accuracy | Tensorflow |
| VOC 2007 Classification [20] | 20 | 2,501 | 4,952 | 11-point mAP | VOC 2007 |
| Oxford-IIIT Pets [66] | 37 | 3,680 | 3,669 | Mean-per-class | Oxford-IIIT Pets |
| GTSRB [80] | 43 | 26,640 | 12,630 | Accuracy | GTSRB website |
| Resisc-45 [13] | 45 | 3,150 | 25,200 | Accuracy | Tensorflow |
| Describable Textures [14] | 47 | 1,880 | 1,880 | Accuracy | DTD website |
| CIFAR-100 [41] | 100 | 50,000 | 10,000 | Accuracy | Tensorflow |
| FGVC Aircraft (variants) [59] | 100 | 3,334 | 3,333 | Mean-per-class | FGVC website |
| Food-101 [8] | 101 | 75,750 | 25,250 | Accuracy | Tensorflow |
| Caltech-101 [22] | 102 | 3,060 | 6,084 | Mean-per-class | Tensorflow |
| Oxford Flowers 102 [65] | 102 | 1,020 | 6,149 | Mean-per-class | Tensorflow |
| Stanford Cars [40] | 196 | 8,144 | 8,041 | Accuracy | Stanford Cars |
| Country-211 [71] | 211 | 31,650 | 21,100 | Accuracy | OpenAI |

Table 5: Statistics of 20 datasets used in image classification.

| Dataset | #Concepts | #Image Train | #Image Test | #Annotated Regions Train | #Annotated Regions Test | Source link |
|---|---|---|---|---|---|---|
| CottontailRabbits | 1 | 1980 | 10 | 2070 | 11 | Roboflow |
| EgoHands(generic) [5] | 1 | 3840 | 480 | 12015 | 1514 | Roboflow |
| Packages | 1 | 19 | 3 | 31 | 5 | Roboflow |
| Raccoon | 1 | 150 | 17 | 164 | 20 | Roboflow |
| Pistols | 1 | 2377 | 297 | 2728 | 358 | Roboflow |
| Pothole | 1 | 465 | 67 | 1256 | 154 | Roboflow |
| NorthAmericaMushrooms | 2 | 41 | 5 | 67 | 9 | Roboflow |
| ThermalDogsAndPeople | 2 | 142 | 20 | 181 | 27 | Roboflow |
| ShellfishOpenImages | 3 | 407 | 58 | 859 | 116 | Roboflow |
| AerialMaritimeDrone(large) | 5 | 52 | 7 | 873 | 78 | Roboflow |
| VehiclesOpenImages | 5 | 878 | 126 | 1676 | 258 | Roboflow |
| Aquarium | 7 | 448 | 63 | 3324 | 584 | Roboflow |
| PascalVOC [20] | 20 | 13690 | 3422 | 31356 | 7835 | Roboflow |

Table 6: Statistics of 13 datasets used in object detection. Box mAP is used as the evaluation metric. Datasets are downloaded from Roboflow. For the datasets without a citation, we refer to Roboflow links for the original sources.

employed (including learning rate and weight decay) to ensure comparison fairness of pre-trained checkpoints, because human-in-the-loop hyper-parameter tuning is excluded in this process. All given training examples are first split into training and validation sets. We perform grid search to select the best hyper-parameter configurations based on the split validation set performance. After it, training is conducted on the entire training set with the selected best hyper-parameters, then the test set performance is reported. Please refer Section 4.1 of [47] for details.

### B.3 More zero-shot results on retrieval

In Table 7, we present additional zero-shot performance of K-LITE on the representative COCO image-text retrieval task. We use COCO 2017 validation set, and show the recall@1 and 5 for image-to-text (I2T) and text-to-image (T2I) retrieval. As shown in the table, K-LITE shows consistent performance improvement over the UniCL baselines with the same amount of pre-training data. This reinforces the generalization ability of K-LITE as we demonstrate in Table 5.

### B.4 How does external knowledge affect visual prediction? Quantitative studies

To further quantitatively study how external knowledge affects the visual prediction, we first identify three factors: $(i)$ Rareness: the inverse frequency of a downstream concept with respect to the pre-training corpus. Specifically, for rareness, we first count the frequency of the concept in the pre-training corpus (#images containing this concept) and the frequency of the concept in the downstream validation set (#images belonging to this concept). We then compute the weighted sum of the frequency, by summing the multiplication result of the two terms. Rareness score of a dataset is

| Training Data | | Method | COCO Retrieval | | | |
| Dataset | # Samples | | I2T R@1 | I2T R@5 | T2I R@1 | T2I R@5 |
|---|---|---|---|---|---|---|
| ImageNet-21K | 13M (full) | UniCL | 2.66 | 7.46 | 0.98 | 3.54 |
| | 13M (full) | K-Lite | **4.04** | **12.20** | **1.91** | **6.48** |
| YFCC-14M + | 14M (half) | UniCL | 21.80 | 45.38 | 13.33 | 32.14 |
| ImageNet-21K | 14M (half) | K-Lite | **22.44** | **47.28** | **14.38** | **33.77** |
| GCC-15M + | 15M (half) | UniCL | 31.88 | 57.76 | 21.41 | 44.69 |
| ImageNet-21K | 15M (half) | K-Lite | **32.68** | **58.88** | **22.08** | **45.41** |

Table 7: Overall comparisons of our knowledge-augmented models on zero-shot COCO Retrieval Evaluation. Each model is pre-trained with 32 epochs following UniCL [95].

computed as the inverse of the weighted sum. A final normalization will be applied so that each number is in the range of 0 and 1. Higher value here denotes higher rareness. $(ii)$ Overlap Difference: Overlap is defined as the percentage of concepts appearing in both training and evaluation; The overlap difference is defined as the absolute overlap improvement after adding external knowledge. $(iii)$ Coverage: The percentage of concepts are covered by the external knowledge bases; We compute the normalized number for each factor in Table 8.

We notice that the largest performance boost comes with Flowers2012 (+30.2), The Flowers2012 dataset has the 7th rarest concepts, high coverage and highest overlap difference (+34%). On the other side, while the StanfordCars dataset has the rarest concerts, the low overlap difference brings only +0.43 performance boost. Furthermore, We computed the Pearson correlation coefficient and found that each factor has a positive correlation while the overlap (especially the difference of the overlap after introducing external knowledge) has the most significant (p=0.001<0.1) correlation with the performance improvement. This suggests that all three (rareness, coverage and overlap) affect the performance of K-Lite, and **the overlap difference plays a major role**.

| Dataset | #Concepts | Improvement | Rareness | Overlap-Difference | Coverage |
|---|---|---|---|---|---|
| Hateful Memes [37] | 2 | 6.64 | 0.33 | 0.00 | 1.00 |
| PatchCamelyon [87] | 2 | 14.00 | 0.33 | 0.00 | 1.00 |
| Rendered-SST2 [71] | 2 | 0.16 | 0.00 | 0.00 | 1.00 |
| KITTI Distance [24] | 4 | 13.30 | 0.01 | 0.01 | 1.00 |
| FER 2013 [1] | 7 | 0.19 | 0.00 | 0.00 | 1.00 |
| CIFAR-10 [41] | 10 | 0.15 | 0.00 | 0.00 | 1.00 |
| EuroSAT [31] | 10 | -2.50 | 0.00 | 0.03 | 1.00 |
| MNIST [16] | 10 | 0.09 | 0.00 | 0.01 | 0.60 |
| VOC 2007 Classification [20] | 20 | -1.60 | 0.02 | 0.00 | 0.85 |
| Oxford-IIIT Pets [66] | 37 | 17.80 | 0.06 | 0.09 | 0.97 |
| GTSRB [80] | 43 | 3.95 | 0.04 | 0.02 | 1.00 |
| Resisc-45 [13] | 45 | 3.29 | 0.03 | 0.06 | 0.98 |
| Describable Textures [14] | 47 | 4.57 | 0.02 | 0.11 | 1.00 |
| CIFAR-100 [41] | 100 | 5.50 | 0.01 | 0.03 | 1.00 |
| FGVC Aircraft (variants) [59] | 100 | -0.10 | 0.00 | 0.01 | 0.10 |
| Food-101 [8] | 101 | 26.40 | 0.01 | 0.21 | 1.00 |
| Caltech-101 [22] | 102 | 1.81 | 0.01 | 0.11 | 1.00 |
| Oxford Flowers 102 [65] | 102 | **30.20** | **0.02** | **0.26** | **0.97** |
| Stanford Cars [40] | 196 | 0.43 | 0.34 | 0.00 | 0.00 |
| Country-211 [71] | 211 | 1.09 | 0.02 | 0.06 | 1.00 |

Table 8: Correlation of the improvement of K-Lite over UniCL on 20 datasets with normalized rareness, overlap difference and coverage.

## B.5 How does external knowledge affect visual prediction? Case studies

**Image Classification.** In Figure 5, we show more examples to illustrate how external knowledge affects visual recognition for the three datasets that knowledge benefit the most. Both success and failure examples are showed.

In Figure 6, we illustrate some cases that the current knowledge quality is low and its improvement is minor or unclear. Improving the knowledge quality can be one interesting research direction to boost the performance for these datasets.

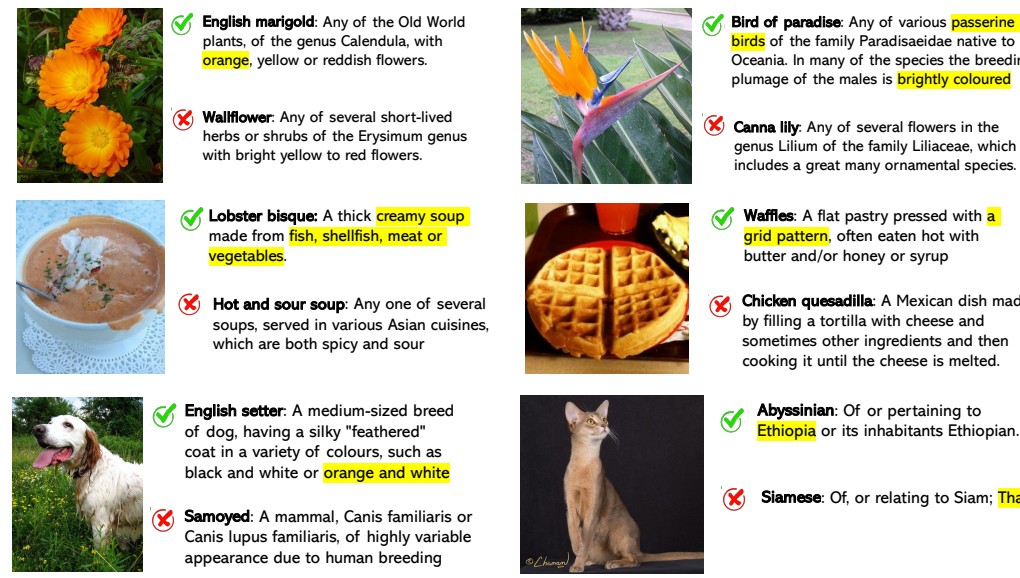

(a) Success examples. The description of the parent concept, material, shape, color *etc.* clarifies the concepts, boosting performance for the fine-grained classification tasks.

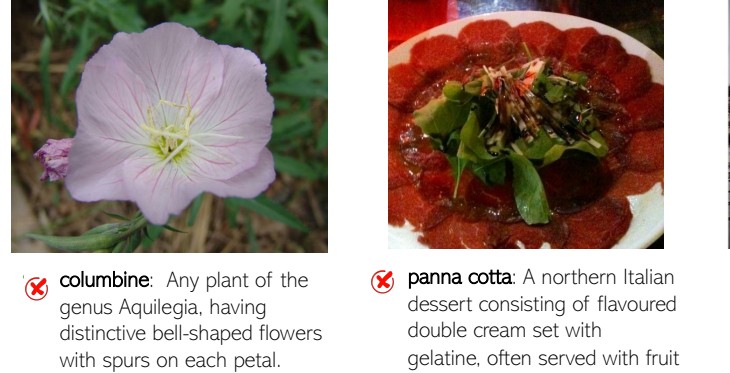
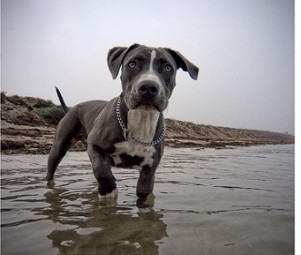

(b) Failure examples. The knowledge contains certain spurious description that confuse the models.

Figure 5: The three datasets with the largest improvement according to Fig. 3: Flowers102, Food101 and OxfordPets. For each dataset, two success examples and one failure example are shown in (a) and (b) respectively. For each image, the top row is the knowledge-based prediction, and the bottom row is the baseline prediction (no knowledge).

**Object Detection.** In Figure 7, we provide examples of object detection results. The pre-training dataset Object365 has concepts "`bread/bun, fire hydrant`", but does not contain concepts "`doughnut, fireplug`". With external knowledge, the unseen concepts "`doughnut, fireplug`" are explained with their shape and similar seen concepts, helping the model to precisely locate the object regions and categorize them into the correct classes with higher confidence. However, similar to IC, there are failure cases due to the fact that external knowledge contains spurious words (the last example in Figure 7), *e.g.,* "`water`" appears in the definition of both "`fireplug`" and "`garden hose`". It confuses the model in the categorization task, though the localization task is better executed.

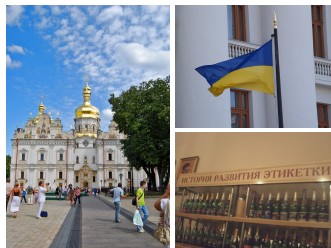

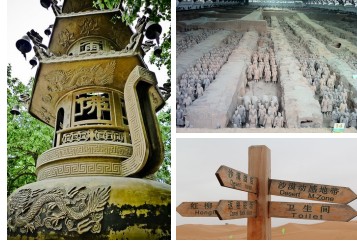

**Ukraine**: A country in Eastern Europe; was long part of the Russian Empire and Austro-Hungarian Empire, then of the Soviet Union.

**China**: a communist nation that covers a vast territory in eastern Asia; the most populous country in the world

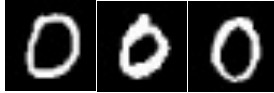

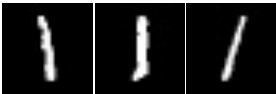

**0**: a mathematical element that when added to another number yields the same number

**1**: the smallest whole number or a numeral representing this number

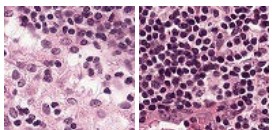

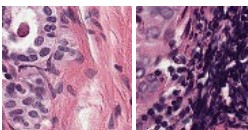

**Lymph node**: Each of the small oval bodies of the lymphatic system, distributed along the lymphatic vessels, that are clustered in the armpits, groin, neck, chest and abdomen. They act as filters, with an internal honeycomb of connective tissue filled with lymphocytes and macrophages that collect and destroy bacteria, viruses and foreign matter from lymph. …

**Lymph node containing metastatic tumor tissue**: Thin, woven, gauze-like fabric.

Figure 6: More case studies for the knowledge-augmented model: Top (Country211), Middle (MNIST), Bottom (PatchCamelyon). External knowledge does not benefit the first 2 datasets significantly (performance gain +1.09% and 0.0%), as the extracted knowledge is not quite relevant to the specific classification task. In PatchCamelyon, the external knowledge improves the baseline with +14.0% absolutely accuracy, we hypothesize the knowledge describes visual appearance of tumor tissue.

## B.6 Adding Grounding Data into Object Detection Training

In the main paper, we focus on pre-training on Object365 dataset. We further train our K-LITE on the combination of Object365 and gold grounding data (GoldG). Follow [50], GoldG is 0.8M human-annotated gold grounding data curated, including Flickr30K, VG Caption, and GQA, while COCO images are removed from the dataset. It is designed to verify the effectiveness of gold grounding data. The zero-shot performance on LVIS is shown in Table 9, where the best numbers of GLIP and K-LITE are reported with their best settings (*e.g.,* adding knowledge or not). By observing more concepts in grounding data, our proposed knowledge-augmented model can achieve superior performance in comparison to the baseline GLIP model. It shows that K-LITE can benefit from further scaling the training data.

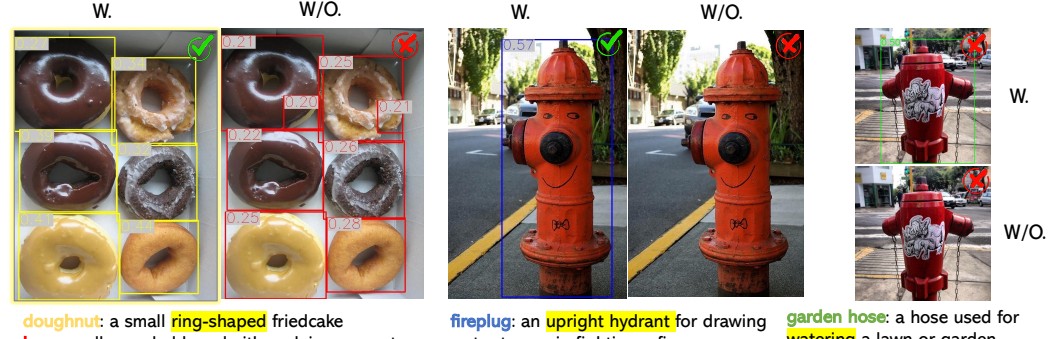

doughnut: a small ring-shaped friedcake
bun: small rounded bread either plain or sweet

fireplug: an upright hydrant for drawing
water to use in fighting a fire

garden hose: a hose used for
watering a lawn or garden

Figure 7: Case study of success (Left & Middle) and failure (Right) for the knowledge-augmented model in object detection. For each image pair, the first one is the knowledge-based prediction (W), and the second one is the baseline prediction without knowledge (W/O). In the inference stage, the prediction box and category above a given threshold is used as the detection result. Only the categories with detected box regions are displayed.

| Method | # Parameters | # Training Data | LVIS | | | |
| --- | --- | --- | --- | --- | --- | --- |
| | | | APr | APc | APf | AP |
| GLIP-A [50] | 151M | Object-365 | 14.2 | 13.9 | 23.4 | 18.5 |
| K-LITE | 151M | | 14.8 | 18.6 | 24.8 | **21.3** |
| GLIP-C [50] | 231M | Object-365 + Gold Grounding Data | 17.7 | 19.5 | 31.0 | 24.9 |
| K-LITE | 151M | | 17.2 | 24.6 | 29.0 | **26.1** |

Table 9: Zero-shot task transfer performance on LVIS dataset. In [50], GLIP-A is a two-encoder model without fusion module trained on Object365, and GLIP-C is a two-encoder model with fusion module trained on Object365 + Gold Grounding Data.