# OpenReview forum: "K-LITE: Learning Transferable Visual Models with External Knowledge"
_NeurIPS.cc/2022/Conference — NeurIPS 2022 Accept_

### Official Review · Reviewer_nX1x · 2022-07-11

**Rating:** 7
**Confidence:** 4
**Soundness:** 3 good
**Presentation:** 3 good
**Contribution:** 3 good

**Summary:**

The paper tackles the poor transfer learning performance of current visual models. It presents a systematic approach to leveraging external knowledge databases to train more transferable and sample-efficient visual models. The paper presents sufficient evidence that external knowledge can benefit large-scale task-level transfer for two core vision problems, image classification, and object detection. The model can achieve similar zero-shot performance to previous methods when using only half of the pre-training image-text pairs in some scenarios.

**Questions:**

Please see the weaknesses.

**Ethics Review Area:**

["I don’t know"]

**Limitations:**

Limitations and potential negative societal impacts are adequately addressed.

**Strengths And Weaknesses:**

Strengths:
1. The motivation is clear. Leveraging external knowledge is promising to increase the concept coverage of vision-language models.
2. Novel solution. Two external knowledge sources including WordNet and Wiktionary are explored to enrich the natural language supervision, which is then used in both the language-image pre-training stage and the prompt-based evaluation stage.
3. Sufficient experiments. The experiments on two downstream tasks with several datasets support the main claims of the paper.

Weaknesses:
1. For knowledge acquisition, when multiple meanings exist for a given query, the method simply considers the first definition. Does the authors analyze the proportion of polysemy and the correct rate of the first-word sense?
2. In order to train more transferable and sample-efficient visual models, the paper instantiates K-LITE with two tasks to demonstrate the generalization ability. Have the authors tested on more tasks, like image-text retrieval?
3. In Table 2, why is the performance of 2-branch lower than 1-branch from scratch?
4. Why not consider other types of external knowledge sources like ConceptNet and knowledge graphs?  What are the advantages of the currently used knowledge sources over others?

---

> ### Author Response · Authors · 2022-08-02
> **Response to Reviewer nX1x**
>
> > For knowledge acquisition, when multiple meanings exist for a given query, the method simply considers the first definition. Does the authors analyze the proportion of polysemy and the correct rate of the first-word sense?
>
> Good question. We simply consider the first definition for the purpose of automating the knowledge acquisition process, and admit that a better definition-choosing strategy can improve the performance. We argue that the most common definition per concept typically comes first in Wiktionary. Since there is no automatic way to quantify the stats at scale, we manually observed around 100 concepts to check this intuition.
>
> ----
>
> > Have the authors tested on more tasks, like image-text retrieval?
>
> Thanks for the good suggestion, we conducted the zero-shot COCO image-text retrieval experiments (with and without external knowledge). We found that K-Lite is strictly better than the UniCL/CLIP counterparts under the same pre-training dataset in Table 3.
>
> - On three pre-training settings: IN21k, IN21k_CC_half, IN21k_YFCC_half
> - For i2t, K-LITE achieves 0.038, 0.327 and 0.224  @R1, while UniCL achieves 0.026, 0.318, 0.218
> - For t2i, K-LITE achieves 0.020, 0.221 and 0.144  @R1, while UniCL achieves 0.009, 0.213, 0.133
>
> This further shows the generalization ability of K-LITE. We will include these results in the revision.
>
> ----
>
> > In Table 2, why is the performance of 2-branch lower than 1-branch from scratch?
>
> Thanks for the insightful observation.
> - For “2-branch, continue pre-training” setting, the performance is lower than “1-branch, from scratch”; we found that it generally requires much more experiment tuning to learn a good knowledge-augmented model, by adapting from a knowledge-free model than training from scratch. It is likely that the best training schedule for “2-branch, continue pre-training” requires more non-trivial experimental exploration. Note that the 2-branch continue pre-training strategy is a fast approach to convert knowledge-free checkpoints to knowledge-augmented ones.
> - For the “2-branch, from scratch” setting, the performance is higher than “1-branch, from scratch”on ImageNet, but lower on “20 datasets”.
> - Compared to the knowledge-free baseline UniCL, both “2-branch” methods improve the performance.
>
> ----
>
> > Why not consider other types of external knowledge sources like ConceptNet and knowledge graphs? What are the advantages of the currently used knowledge sources over others?
>
> Due to limited bandwidth, we focus on two commonly used knowledge bases, WordNet and Wiktionary. We believe it is a good idea to explore more comprehensive external knowledge sources like ConceptNet (note, much of ConceptNet knowledge comes from Wiktionary). We leave this as future work.
>
> We explored full Wikipedia pages the beginning of our experiments, but found it is hard to identify the proper knowledge sequence from the long documents. Thus, one advantage of WordNet & Wiktionary is its simplicity to obtain the concise knowledge.
>
> ----

---

### Official Review · Reviewer_vBeu · 2022-07-11

**Rating:** 7
**Confidence:** 3
**Soundness:** 3 good
**Presentation:** 3 good
**Contribution:** 3 good

**Summary:**

This paper proposes a knowledge base augmented image-language model. The main architecture is an image-text contrastive training like CLIP. The key idea is for the text input, a "keyword" (a noun) is extracted and used as a query to search in multiple knowledge bases to get the word hierarchy or definition. This additional textual information is also input to the image-language model as a language prompt. The model is evaluated on multiple downstream datasets on image classification and object detection tasks. Results show incorporating an external knowledge base is effective.


**Questions:**

1. In Table 3, the bold number 21.3 is confusing. There is a 21.4 in the same row under LVIS tab. What's the meaning of bold font here?
2. In the query construction section (L144) the paper chooses a single noun as a query to search in the knowledge base. Are there more discussions on this design choice? For example, if it's helpful to query `adj+noun` for a more accurate definition, or sometimes `verb+noun` if the text input describes an action.


**Limitations:**

There is no major negative societal impact of the work. The paper also briefly discussed it in the appendix.


**Strengths And Weaknesses:**

The paper acknowledges related works (L111) that incorporate knowledge base in the NLP or vision-language models. The paper focuses on the transfer ability (transfer to specialized datasets and rare objects) of such a knowledge base augmented model, which is novel in the field.

The quality of the paper is very good. The ablation studies are well conducted. The experimental results are strong.

In general, the paper is clearly written and well organized.

The experimental results show the proposed method works well on a wide range of specialized datasets for both image classification and object detection tasks. This method of incorporating a knowledge base can be applied to other research.

---

> ### Author Response · Authors · 2022-08-02
> **Response to Reviewer vBeu**
>
> > In Table 3, the bold number 21.3 is confusing. There is a 21.4 in the same row under LVIS tab. What's the meaning of bold font here?
>
> Sorry for the typo. We will make 21.4 bold.
>
> ----
>
> > Are there more discussions on this design choice? For example, if it's helpful to query adj+noun for a more accurate definition, or sometimes verb+noun if the text input describes an action.
>
> We thank the reviewer for this suggestion. In our preliminary experiments, we ablated different ways to construct queries on CC3M (including entity, noun phrase, noun, etc). Among these choices, we found that the noun phrase works the best. We will detail our design choice in the final version.
>
> ----

---

> > ### Comment · Reviewer_vBeu · 2022-08-08
> > **Reply to Authors**
> >
> > Thanks for the response. I keep my rating as an "Accept".

---

### Official Review · Reviewer_AT54 · 2022-07-13

**Rating:** 6
**Confidence:** 4
**Soundness:** 3 good
**Presentation:** 4 excellent
**Contribution:** 2 fair

**Summary:**

The paper proposes K-LITE which leverages external knowledge as a more structured source of supervision signals for building transferable visual models with knowledge-augmented language-image pre-training. External knowledge can enrich the natural language supervision by explaining rare concepts with more common concepts thus increasing the concept coverage. K-LITE proposes systematic steps to construct queries to get knowledge from external sources such as WordNet or Wiktionary to facilitate the pre-training and evaluation for image classification and object detection tasks. To bridge the gap between pre-training and evaluation, the paper utilizes adapters to further modularize the original language encoder with external knowledge data. With extensive experiments conducted, the paper demonstrates K-LITE’s great generality for downstream datasets and data effectiveness on both image classification and object detection tasks.

**Questions:**

1. In Section 3.1, how to compute the rare level of a noun phrase? Note that in Line 153, the paper said that the most *rare* noun-phrase over the corpus is used as a query for the sentence.
2. For the example sentence in Line 156, will all the three noun phrases “boxer”, “professional boxer”, and “the crowd” be used as queries? If the answer is yes, this is not aligned with “the most *rare* noun-phrase” in Line 153.
3. Typo in Line 279: “therefor” should be “therefore”, “mudularized” should be “modularized”.

**Limitations:**

The paper has reported limitations and offered satisfactory explanations. No obvious negative societal impact was found.

**Strengths And Weaknesses:**

Strengths:
1. The paper is well-written and easy to follow. The motivation of leveraging external knowledge to help language-image pre-training is natural and clear.
2. With knowledge-augmented training and evaluation, K-LITE achieves superior task-level transfer performances on 20 image classification and 13 object downstream tasks. In addition, K-LITE can get comparable performance as UniCL with only half of the pre-training data, showing great data efficiency.
3. The paper provides detailed and persuasive analyses and discussions to confirm the design choices of K-LITE. The analysis of train-evaluation concept overlap (increased from 13.26% to 51.24% with knowledge) well explained why external knowledge can facilitate language-image pre-training and task-level transfer to downstream IC and OD tasks.

Weaknesses:
1. While the motivation is natural, it is not surprising. Since many prior works have leveraged external knowledge for language-only and vision-language tasks before. Even for know-augmented visual recognition, there is at least a similar idea proposed [1], but the K-LITE paper didn’t cite [1]. Therefore, the technical novelty seems relatively limited.  *[1] Tian C, Wang W, Zhu X, et al. VL-LTR: Learning Class-wise Visual-Linguistic Representation for Long-Tailed Visual Recognition[J]. arXiv preprint arXiv:2111.13579, 2021.*

2. Table 1 reveals that training/test inconsistency in terms of involving knowledge can dramatically degrade the model performance. Thus, to ensure higher performance, during evaluation, detailed explanations for noun phrases always need to be concatenated as knowledge prompts. This seems to be not aligned with the motivation of the paper and the spirit of utilizing external knowledge. For example, if we already what is “sashimi”, we don’t need to find the definition every time when we see the word later. I think this paper didn’t solve this motivation-reality gap very well. Even though applying K-Adapter can alleviate the problem, it didn’t bring any improvement in terms of accuracy compared with knowledge-free pre-training according to Table 1.
3. Since detailed explanations are appended to noun phrases, the length of the input sequence will be much longer than the previous knowledge-free category name or caption. This will definitively result in longer inference time and larger training costs. Although K-LITE enjoys better data efficiency, the author should also provide the training and inference budgets to show whether K-LITE possesses another aspect of efficiency.
4. From Figure 4 in the main paper and Figure 5 in the appendix, we can find that the performance and improvement of K-LITE largely depend on the quality of the external knowledge base. The method itself cannot guarantee an improved performance at the cost of external organized/structured data under some circumstances (just as the Limitations subsubsection stated).

---

> ### Author Response · Authors · 2022-08-02
> **Response to Reviewer AT54**
>
> > While the motivation is natural, it is not surprising. Since many prior works have leveraged external knowledge for language-only and vision-language tasks before. Even for know-augmented visual recognition, there is at least a similar idea proposed [1], but the K-LITE paper didn’t cite [1]. Therefore, the technical novelty seems relatively limited. [1] Tian C, Wang W, Zhu X, et al. VL-LTR: Learning Class-wise Visual-Linguistic Representation for Long-Tailed Visual Recognition[J]. arXiv preprint arXiv:2111.13579, 2021.
>
> Thanks for the reference, we are happy to discuss the paper in related work. Indeed, Wiki knowledge is leveraged in VL-LRT to improve long-tail concept recognition. Note that VL-LRT is still in the category of class-level transfer (training & evaluation in the same domain / dataset), while we focus on task-level transfer (pre-training on large corpus, and transfer to a wide range of downstream datasets in different domains).
>
> ----
>
> > Table 1 reveals that training/test inconsistency in terms of involving knowledge can dramatically degrade the model performance. Thus, to ensure higher performance, during evaluation, detailed explanations for noun phrases always need to be concatenated as knowledge prompts. This seems to be not aligned with the motivation of the paper and the spirit of utilizing external knowledge. For example, if we already what is “sashimi”, we don’t need to find the definition every time when we see the word later. I think this paper didn’t solve this motivation-reality gap very well. Even though applying K-Adapter can alleviate the problem, it didn’t bring any improvement in terms of accuracy compared with knowledge-free pre-training according to Table 1.
>
> It is not harmful to add external knowledge for common concepts, except for the efficiency concern; The external knowledge here is a necessary link to bridge rare & common concepts, This is also validated in other NLP paper [*].
>
> We would like to kindly point out that the comment is incorrect "*Even though applying K-Adapter can alleviate the problem, it didn’t bring any improvement in terms of accuracy compared with knowledge-free pre-training according to Table 1.*" The knowledge-free baseline is UniCL in the 1st row of Table1, which yields 28.16 on ImageNet-1K and 27.15 on 20 datasets. Compared with this, both 2-branch methods (continue pre-training & from scratch) improve the performance.
>
>  [*] Wang, Xiaozhi, et al. "KEPLER: A unified model for knowledge embedding and pre-trained language representation." TACL 2021.
>
> ----
>
> > Since detailed explanations are appended to noun phrases, the length of the input sequence will be much longer than the previous knowledge-free category name or caption. This will definitively result in longer inference time and larger training costs. Although K-LITE enjoys better data efficiency, the author should also provide the training and inference budgets to show whether K-LITE possesses another aspect of efficiency.
>
> In terms of training & inference efficiency, KLITE is generally the same asthe baseline UniCL/CLIP. Specifically, each epoch takes 1.14hr for K-LITE and 1.13hr for UniCL/CLIP on CC15M though we introduce external knowledge text. This is due to the fact of token padding in implementation. Following the implementation details of OpenAI CLIP where the max sequence length is set to 77, padding tokens will be added and processed if a sequence length is shorter than 77. In K-LITE, we also set max length as 77. Besides the original language sequence, the additional knowledge sequence tokens will replace the redundant padding tokens under the 77 length constraint.
>
> In preliminary experiments, we also tried to increase the max sequence length to 128 to accommodate long knowledge texts. This would lead to around 2% performance improvement but with 46% compute overhead. We will point out the fact in our final version.
>
> ----
>
> > From Figure 4 in the main paper and Figure 5 in the appendix, we can find that the performance and improvement of K-LITE largely depend on the quality of the external knowledge base. The method itself cannot guarantee an improved performance at the cost of external organized/structured data under some circumstances (just as the Limitations subsubsection stated).
>
> As the reviewer pointed out, it is right that the performance and improvement of K-LITE largely depend on the quality of the external knowledge base. We would like to emphasize that the issue is an open problem for all the knowledge-augmented methods in NLP, vision-and-language, and now in computer vision (as shown in our paper). Improving the quality of knowledge bases is another important and active research topic.

---

> > ### Author Response · Authors · 2022-08-02
> > **Response to Questions**
> >
> > > In Section 3.1, how to compute the rare level of a noun phrase? Note that in Line 153, the paper said that the most rare noun-phrase over the corpus is used as a query for the sentence.
> >
> > For a pre-training dataset, we compute the frequency of each noun phrases appearing in the dataset. The rare noun-phrases are considered as the those lower frequency.
> >
> > ----
> >
> > > For the example sentence in Line 156, will all the three noun phrases “boxer”, “professional boxer”, and “the crowd” be used as queries? If the answer is yes, this is not aligned with “the most rare noun-phrase” in Line 153.
> >
> > No, only the most rare noun-phrase is consider (ie, the lowest frequnency). In this example, only “professional boxer” is chosen.
> >
> > ----
> >
> > > Typo in Line 279: “therefor” should be “therefore”, “mudularized” should be “modularized”.
> >
> > Thanks. We will correct them.

---

> > > ### Comment · Reviewer_AT54 · 2022-08-08
> > > **Thanks for the response**
> > >
> > > The detailed response from the authors solved most of my concerns. After reading the author response and other review comments, I decided to keep my original rating as "Weak Accept".

---

### Official Review · Reviewer_Q5MT · 2022-07-15

**Rating:** 7
**Confidence:** 4
**Soundness:** 3 good
**Presentation:** 3 good
**Contribution:** 3 good

**Summary:**

This paper proposes to use external knowledge for learning multi-modal representation from text and images. The authors extract a query given a text description attached to an image. They use the query to extract more informative descriptions from the freely available external knowledge. The externally obtained textual description is then paired with the image to enhance the jointly learned visual representation.

**Questions:**

As I mentioned above, the authors could perform some interesting experiments to elaborate more on performance. For instance, which classes benefit more from exploring external knowledge? Can it be somehow quantified based on a rareness score? According to the main idea proposed in the introduction, the rare classes should benefit more from external knowledge. You already have a measure to determine the rareness of noun phrases used in the query extracting process. Can you use the same measure to see if there is a correlation between the rareness of a given class and the obtained boost after using external knowledge?

**Limitations:**

Yes, the authors adequately addressed the limitations and potential negative societal impact of their work.

**Strengths And Weaknesses:**

This paper's idea is simple, effective, and properly presented. The paper is technically sound, and the experiments match NeurIPS standards.

The only weakness I see in this paper is that the authors are content with reporting the final performance to validate the proposed ideas. They show more detailed analyses in the appendix. However, I believe conducting a set of ablation studies will strengthen the paper. See below for more details.

---

> ### Author Response · Authors · 2022-08-02
> **Response to Reviewer Q5MT**
>
> > As I mentioned above, the authors could perform some interesting experiments to elaborate more on performance. For instance, which classes benefit more from exploring external knowledge? Can it be somehow quantified based on a rareness score? According to the main idea proposed in the introduction, the rare classes should benefit more from external knowledge. You already have a measure to determine the rareness of noun phrases used in the query extracting process. Can you use the same measure to see if there is a correlation between the rareness of a given class and the obtained boost after using external knowledge?
>
> We thank the reviewer for asking this insightful question. We agree with the reviewer that more insights and analysis can benefit future research, and will briefly discuss why external knowledge improves transfer learning immediately below. We first identify three factors:
> - Rareness: the frequency of a downstream concept wrt the pre-training corpus.
> - Coverage: a concept is covered by the external knowledge bases or not;
> - Overlap: the percentage of concepts appearing in both training and evaluation.
>
> We further computed the Pearson correlation coefficient and found that each factor has a positive correlation while the overlap (especially the difference of the overlap after introducing external knowledge) has the most significant (p=0.001<0.1) correlation with the performance improvement. Qualitatively, we found that the largest performance boost comes with Flowers2012 (+30.2), The Flowers2012 dataset has the 18th rarest concepts, high coverage and highest overlap difference (+34%). On the other side, while the StanfordCars dataset has the rarest concerts, the low overlap difference brings only +0.43 performance boost. This suggests that all three (rareness, coverage and overlap) affect the performance of K-Lite and the overlap difference plays a major role.
>
> ----

---

### Author Response · Authors · 2022-08-02
**Summary of Rebuttal**

We sincerely thank all the reviewers for their time and their thoughtful comments and questions. We are encouraged that the reviewers find our method novel (uUn7, Tnhg, RnhL), simple yet effective (uUn7, TGns, sC7n) and generalizable (uUn7, Tnhg), and that the paper is well-presented (sC7n, RnhL) and contains extensive experiments (sC7n, RnhL).

We attempted our best to address the questions as time allowed. We believe the comments & revisions have made the paper stronger and thank all the reviewers for their help. Please find individual responses to your questions below.  More response with quantitive results will be provided once available.

---

### Author Response · Authors · 2022-08-09
**Paper Updated**

We have revised our paper based on the suggestions from reviewers, with the main changes marked in blue. Here is the summary of major updates:

- (Reviewer nX1x) In Section B.3, we provide zero-shot retrieval comparisons by introducing knowledge.
- (Reviewer Q5MT) In Section B.4, we provide quantitative analysis on how external knowledge benefits transfer, including rareness, overlap, coverage
- (Reviewer nX1x, AT54)  We re-evaluate our checkpoint, and update the results of `2-branch, from scratch` in Table 1. The performance of 2-branch is consistently higher than 1-branch from scratch.

---

### Meta-Review · Area_Chair_vGwN · 2022-08-26

**Recommendation:** Accept
**Confidence:** Certain

**Metareview:**

This paper presents a simple yet effective technique for leveraging external knowledge to learn multi-modal representation. The performance on image classification and zero-shot object detection dataset is surprisingly great. All reviewers give positive comments. After the discussion phase, the authors revise the paper accordingly and provide zero-shot retrieval comparisons, more quantitative analysis, reevaluate the checkpoint. The rebuttal satisfied the reviewers. The meta-reviewers thus recommend accepting it.

**Award:**

No

---

### Decision · Program_Chairs · 2022-09-14

Accept